# MOUSTERIAN: EXPLORING THE EQUIVALENCE OF GENERATIVE AND REAL DATA AUGMENTATION IN CLASSIFICATION

## ABSTRACT

In this paper, we address a key question in machine learning: **How effectively can generative data augmentation enhance image classification?** We begin by examining the differences and similarities between real and synthetic data generated by advanced text-to-image models. Through comprehensive experiments, we provide systematic insights into leveraging synthetic data for improved classification performance. Our findings show that: 1). Generative data augmentation by models trained solely on the internal (available training) set can effectively improve classification performance, validating the long-held hypothesis that synthesis enhances analysis by enriching modeling capability. 2). For generative data augmentation by models trained on both internal and external data (e.g. large-scale image-text pairs) separately, the size of equivalent synthetic dataset augmentation can be determined empirically. In addition to being aligned with a common intuition that real data augmentation is always preferred, our empirical formulation also provides a guideline for quantitatively estimating how much larger the size of generative dataset augmentation is, over the real data augmentation, to achieve comparable improvements. Our CIFAR-10 and ImageNet results also demonstrate its impact w.r.t. the size of the baseline training set and the quality of generative models.

## 1 INTRODUCTION

We assume the task of predicting labeling $y$ for a given input $\mathbf{x}$. The *analysis-by-synthesis* methodology (Yuille & Kersten, 2006) has once been considered as one of the guiding principles for making a variety of inferences (Cootes et al., 1995; Tu & Zhu, 2002; Fergus et al., 2003). The school of thought in pattern theory (Grenander, 1993) considers the capability of being able to synthesize (being generative) stands at the utmost important position for making robust, transparent, and effective analysis/inference. The analysis-by-synthesis principle would also expect having powerful generative $p(\mathbf{x}|y)$ (*e.g.* text-to-image generation (Ramesh et al., 2021; Rombach et al., 2022)) can substantially improve the inference of $p(y|\mathbf{x})$. For image classification, one would expect that adding synthesized images to datasets like ImageNet (Deng et al., 2009b) as data augmentation would lead to an immediate improvement. However, the view of analysis-by-synthesis for visual inference has been challenged in the big data and deep learning era (Goodfellow et al., 2016; LeCun et al., 2015).

For the sake of clarity, we define *synthetic* data here as images generated by statistical generative models, distinguishing them from 'synthetic' data produced by graphics simulation engines (Beery et al., 2020).

There is an explosive development with increasing level of maturity in image generation, including generative adversarial learning (Tu, 2007; Goodfellow et al., 2014; Karras et al., 2018), variational autoencoder (VAE) (Kingma, 2013), and diffusion models (Sohl-Dickstein et al., 2015; Ho et al., 2020a; Rombach et al., 2022; Ramesh et al., 2021). With the increasing representation power and photo-realism of generative modeling, especially diffusion-based models, we make a timely effort to answer the question about the *effectiveness of generative data augmentation for image classification*.

Previous attempts exist to partially address the above question. For instance, studies such as (Azizi et al., 2023; Fan et al., 2024) demonstrate that the ImageNet classification accuracy can be improved by incorporating synthetic data generated by state-of-the-art generative models, which are pre-trained

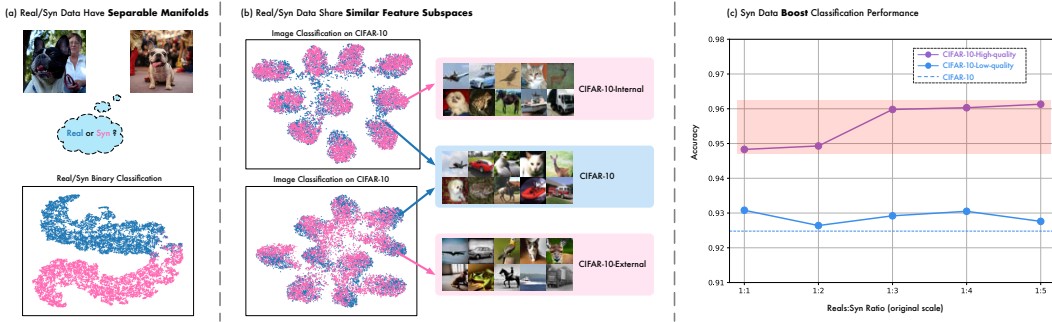

Figure 1: **The differences and equivalences between real and synthetic data**: (a) **Manifold Distinctions**: The manifolds of the real and the synthetic data in subspaces learned by a binary domain classifier, highlighting their significant domain gap. (b) **Feature Subspace Overlap**: The top row shows manifolds for CIFAR-10 and CIFAR-10-Internal (a synthetic dataset generated by Vanilla-DDPM trained solely on CIFAR-10 itself). The bottom row shows similar results for CIFAR-10-External (the CIFAKE dataset (Bird & Lotfi, 2024) generated by Stable Diffusion 1.4 trained on a subset of LAION-5B). Both figures reveal notable overlap in feature subspaces. (c) **Performance Gains**: Augmenting the real training set with high-quality synthetic data leads to evident improvements in classification performance.

on large-scale external data and subsequently fine-tuned on the target dataset. 1). Different from all prior works (Azizi et al., 2023; Fan et al., 2024) that adopt generative data augmentation by models trained on external data, we start our investigation from the very basic problem setting of image classification by studying generative data augmentation from models trained solely on the **internal** (given training) set. 2). Next, we provide **quantifiable guidance** regarding the size of generative data augmentation by **internal** and **external** data. Both aspects have been under-explored in the past.

Suppose we are given a set of training data $\mathbf{S}_{\text{base}} = \{(\mathbf{x}_i, y_i), i = 1, .., n_{\text{base}}\}$, where $\mathbf{x}_i$ indexes the $ith$ training image with its corresponding ground-truth label $y_i$. Let $\mathbf{S}_{\text{syn}}^{+} = \{(\mathbf{x}_j', y_j), j = n_{\text{base}} + 1, .., n_{\text{base}} + n_{\text{syn+}}\}$ be an augmented training set of synthesized images where $\mathbf{x}_j'$ refers to each synthesized image; let $\mathbf{S}_{\text{real}}^{+} = \{(\mathbf{x}_j, y_j), j = n_{\text{base}} + 1, .., n_{\text{base}} + n_{\text{real+}}\}$ be an augmented set of real images.

We present our work, **Mousterian**, via a comprehensive study to derive key

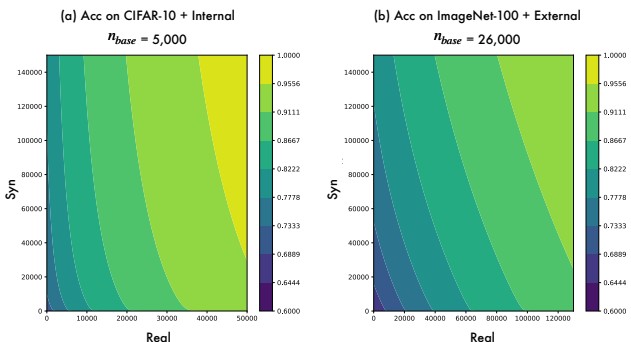

Figure 2: **Equivalence curves** regarding the amount of additional real data and synthetic data at fixed $n_{\text{base}}$ under both internal and external settings.

findings that offer systematic guidance on effectively leveraging synthetic data to boost classification performance. We report the following new findings:

- Generative data augmentation by models trained solely on the available training (**internal**) set can effectively boost classification performance, validating the long-held hypothesis that synthesis enhances analysis by enriching modeling capability.

- Given a training set $\mathbf{S}_{\text{base}}$ (on the CIFAR-10 set) together with a generative model trained on the $\mathbf{S}_{\text{base}}$ (**internal**), $G_{internal}(\cdot)$[1], we obtain an empirical equivalence for the generative data augmentation size $|\mathbf{S}_{\text{syn}}^{+}| = n_{\text{syn+}}$ w.r.t. the real data augmentation size $|\mathbf{S}_{\text{real}}^{+}| = n_{\text{real+}}$ as:

$$\frac{n_{\text{syn+}}}{n_{\text{base}}} \simeq 0.6 \times 9.8^{\frac{n_{\text{base}}}{15571.2}} \times \left(1.3^{\frac{n_{\text{real+}}}{0.2 n_{\text{base}}}} - 1\right). \quad (1)$$

- On the ImageNet dataset, let $G_{external}(\cdot)$ denote a cutting-edge diffusion model that is trained on large-scale text-image pairs (**external**), the data augmentation size $|\mathbf{S}_{\text{syn}}^{+}| = n_{\text{syn+}}$ w.r.t. the

---

[1]with the abuse of certain specific variations regarding e.g. the quality of the generative models

real data augmentation size $|\mathbf{S}_{\text{real}}^{+}| = n_{\text{real+}}$ can be determined as:

$$\frac{n_{\text{syn+}}}{n_{\text{base}}} \simeq 4.0 \times 1.3^{\frac{n_{\text{base}}}{15571.2}} \times \left(1.1^{\frac{n_{\text{real+}}}{0.3 n_{\text{base}}}} - 1\right). \tag{2}$$

Figure 2 shows the corresponding equivalence curves that give rise to Eq. 1 and Eq. 2. Although Eq. 1 and Eq. 2 are not directly comparable (the classification settings and generative models are different), it is nevertheless evident that for both internal and external generative data augmentation: 1). To achieve the same level of a performance boost, the required **synthetic data size is always greater than that of the real data**, meaning that having the real data is always more advantageous than using synthetic data; 2). The required generative data augmentation **goes up when the base training set increases**, meaning that it is more challenging to improve the performance when the basic classification accuracy is already strong. Note that both Eq. 1 and Eq. 2 are meant to serve as an empirical quantification for high-level guidance. To the best of our knowledge, this is the first work of its kind allowing us to see the quantitative equivalence of using real vs. generative data augmentation for image classification.

## 2 MOTIVATION

In this section, we will present initial observations of synthetic data, discuss the algorithms and strategies we have tried based on these observations, and summarize our main conclusions.

### 2.1 INTERNAL GENERATIVE DATA AUGMENTATION

We begin by considering the internal setting, where synthetic data used for augmentation is generated by models trained exclusively on the given training set. In this scenario, any observed distribution gap between the real and synthetic data might be attributed to **inherent limitations in the generative model itself**. To validate the existence of this gap, we employ a straightforward method to highlight the differences between the real and synthetic data distributions. Specifically, we train a ResNet-101 model (He et al., 2016) to classify between images from these two domains on CIFAR-10. Our results demonstrate that the binary classification accuracy using high-quality and low-quality internal synthetic data both exceed **98%**. Ideally, a generative model should produce synthetic data that is statistically indistinguishable from real data, capturing the full complexity and diversity of the dataset. However, these findings suggest that, due to inherent limitations of the generative model, real and synthetic data exist on distinct manifolds, even though synthetic data may sometimes appear visually realistic. Basically, the space of all valid images is immensely large and any existing generative models can only cover a small subspace of the sampling space, resulting a fundamental differences in the statistics of the image patches between synthetic and real. To further validate the distributional differences, we visualize the feature vectors from real and synthetic domains, as shown in Figure 1 (a), where the distribution gap is clearly significant. Additional experiments and analyses are provided in Appendix C.1.

### 2.2 EXTERNAL GENERATIVE DATA AUGMENTATION

A more common approach is to use generative models pre-trained on large external datasets. While these models can enhance image accuracy and diversity, they introduce another gap: **the difference between the external dataset and the given classification dataset**. When combined with the inherent limitations of the generative model mentioned earlier, the resulting synthetic data shows an even greater distributional disparity from the given real data. Figure 1 (b) presents the manifolds of CIFAR-10 and synthetic datasets extracted by a standard image classifier trained solely on CIFAR-10. The CIFAR-10-Internal (top row) refers to the synthetic dataset generated by a Vanilla-DDPM trained only on CIFAR-10, which is the given data. The CIFAR-10-External (bottom row) represents the CIFAKE dataset (Bird & Lotfi, 2024), generated by Stable Diffusion 1.4 (Rombach et al., 2022), which is trained on a subset of LAION-5B (Schuhmann et al., 2022), thus incorporating external data. While both synthetic datasets show overlaps with the real data on certain feature projections, the overlap between CIFAR-10-Internal and CIFAR-10 is notably more pronounced.

### 2.3 CAN SYNTHESIZED DATA HELP WITH CLASSIFICATION?

Significant overlapping in Figure 1 (b) raises an important question: Can synthetic data improve classification performance? To explore this, we conduct experiments on CIFAR-10 with two synthetic datasets of different quality. As for relatively low-quality data synthesis, We utilize the synthetic

dataset generated by Vanilla-DDPM (as previously mentioned), and for high-quality data, we employ a diffusion model pre-trained on CIFAR-10 that utilizes advanced training and sampling techniques introduced in EDM (Karras et al., 2022) to generate high-quality data. A detailed comparison of data quality is provided in Section 3.1. Next, we augment the CIFAR-10 training set with synthetic data at different ratios. As shown in Figure 1 (c), augmenting the real training set with high-quality synthetic data results in evident improvements in classification performance. Based on this observation, we adopt a mixed training strategy to address a fundamental question in image classification: Given a dataset comprising a certain quantity of real images, how can generative models effectively enhance classifier performance? Our extensive experiments lead to the following key conclusions, which hold true for **both the internal and external settings**. We will now present these conclusions and explain how they align with the formula we have proposed in Section 1.

- *Through the mixed training strategy, synthetic data can improve the performance of classification, especially when the real data is limited.*

  Let the ratio of added synthetic data $\frac{n_{\text{syn+}}}{n_{\text{base}}}$ be denoted as $r_{\text{syn+}}$, and the ratio of added real data $\frac{n_{\text{real+}}}{n_{\text{base}}}$ be denoted as $r_{\text{real+}}$. As shown in Eq. 1 and Eq. 2, when $r_{\text{real+}}$ is fixed, a decrease in $n_{\text{base}}$ corresponds to a decline in $r_{\text{syn+}}$, demonstrating that the benefits of synthetic data are more pronounced when real data is scarce.

- *Synthetic data tends to be less sample-efficient than real data, with a single real data point equivalent to multiple synthetic copies.*

  Regardless of the fixed value of $n_{\text{base}}$, the calculated value of $n_{\text{syn+}}$ always exceeds $n_{\text{real+}}$, suggesting that the necessary increase in synthetic data to achieve a comparable performance boost is greater than that required for real data. This conclusion can also be observed in the contour plot in Figure 2, where the intersection points of each contour line with the vertical axis are consistently higher than those with the horizontal axis.

- *The performance gains from synthetic data diminish rapidly as the amount of synthetic data increases.*

  From Eq. 1 and Eq. 2, we can see that when $n_{\text{base}}$ is fixed, $r_{\text{syn+}}$ grows exponentially with $r_{\text{real+}}$, which indicates that when more synthetic data is added, its effectiveness will indeed reach saturation.

## 3 SHAPING THE CLASSIFICATION MODELS WITH SYNTHETIC DATA BY GENERATION

In this section, we will outline the experiments conducted to address the problem of how to use generative models for data augmentation, accompanied by a comprehensive and various set of conclusions. It is worth noting that the key conclusions mentioned in the last section will be revisited later from a new experimental perspective. We first explore the effectiveness of internal and external generative data augmentation. Subsequently, under the condition of mixed training with real and synthetic data, we examine how factors such as the quality of $\mathbf{S}_{\text{syn}}^{+}$, the mixing ratio $n_{base} : n_{syn+}$, and the size of real data $n_{base}$ influence improvements in classification performance.

### 3.1 GENERATIVE MODELS TRAINED ON **INTERNAL** DATA FOR CIFAR-10 CLASSIFICATION

When we have a real dataset to train a classifier, two approaches for generating synthetic data naturally come to mind: first, training a generative model solely on the **internal** data (given dataset); second, using an off-the-shelf generative model pre-trained on a large **external** dataset. Whether synthetic data created by generative models trained on internal data can help boost performance is underexplored in previous studies. Here, we investigate this scenario with the CIFAR-10 dataset (Krizhevsky et al., 2009).

**Experiment Setup**  To conduct our experiments, we work with the full CIFAR-10 training set (5,000 samples per class), as well as subsets referred to as **CIFAR-Half** (2,500 samples per class) and **CIFAR-Small** (500 samples per class). We utilize a Vanilla-DDPM trained solely on CIFAR-10 to generate synthetic data, mixing it with real data at different ratios during training. Unless specified otherwise, we use a ResNet-110 model as our classification backbone. All evaluations are done on the original CIFAR-10 validation set.

**Empirical equivalence**    We explore how many additional synthetic samples $n_{\text{syn+}}$ are equivalent to a given amount of additional real data $n_{\text{real+}}$ under a fixed $n_{\text{base}}$ using the data points of Vanilla-DDPM. To be specific, we first fit the relationship between classification accuracy and the variables $n_{\text{base}}$, $n_{\text{real+}}$, and $n_{\text{syn+}}$. The resulting contour plot at fixed $n_{\text{base}}$ of 5,000 is shown in Figure 2 (a). Using the fitted accuracy function, we then determine the synthetic data amount $n_{\text{syn+}}$ that achieves the same accuracy as $n_{\text{real+}}$ added real samples, leading to the derivation of the formula in Eq. 1. The data used for fitting is provided in Appendix C.2.

**Main Results**    *Even with a limited amount of real data, training a generative model on it can still improve classifier performance.*

Collecting real data at scale can often be challenging. In this section, we explore whether generative models can enhance classification performance when the amount of the given real data is relatively limited and no external data is utilized. We train a generative model on the CIFAR-Small dataset and use it to generate synthetic data, varying from one to four times the size of the corresponding real dataset. For comparison, we also conduct similar experiments using the same generation protocol on CIFAR-Half and the entire CIFAR-10 dataset.

Surprisingly, despite the very low quality of the generated images on CIFAR-Small (see Figure 14 in Appendix for visualization), the improvement achieved with these synthetic data is even more pronounced than that observed with CIFAR-Half and the full CIFAR-10 dataset, as shown in Figure 3. We hypothesize that this is due to the strong inductive bias

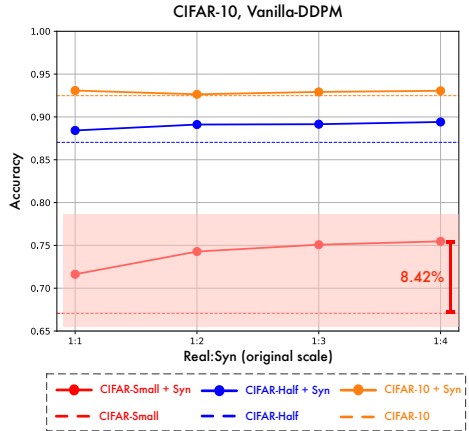

Figure 3: **Comparison of accuracy** using synthetic data generated from CIFAR-Small, CIFAR-Half, and the full CIFAR-10 dataset.

of generative models. With a small amount of real data, although the generative model may not produce perfectly accurate images, it can still capture key features such as the shape of an airplane or the texture of a frog's skin. This capability significantly enhances the generalization performance of discriminative models when data is scarce.

### 3.2    GENERATIVE MODELS TRAINED ON **EXTERNAL** DATA FOR CIFAR-10 CLASSIFICATION

A prevalent approach for data augmentation is to utilize generative models that have been pre-trained on large external datasets. While these models can provide the classifier with **external knowledge**, as it has been exposed to a large amount of data unseen by the classifier, they also introduce an additional challenge mentioned in Section 2—the **disparity** between the external dataset and the specified classification dataset. This raises important a question: Is it beneficial to train a generative model on external data? To explore this question, we conduct the following experiments.

**Experiment Setup**    Using the full CIFAR-10 training set as the real training data, we consider three methods for generating synthetic data: 1). Optimize a conditional Vanilla-DDPM (Ho et al., 2020b) on CIFAR-10 from scratch. 2). Use a diffusion model pre-trained on CIFAR-10 that employs advanced training and sampling techniques introduced in EDM (Karras et al., 2022) to generate higher-quality synthetic images. 3). Utilize the synthetic data in CIFAKE dataset (Bird & Lotfi, 2024), which is generated by Stable Diffusion 1.4 (Rombach et al., 2022) trained on a subset of LAION-5B (Schuhmann et al., 2022), thus incorporating substantial external knowledge.

**Main Results**    *External generative data augmentation is useful, but even without it, using cutting-edge generative models trained on internal data still has the potential to improve classification.*

As shown in Table 1, CIFAKE has a much higher FID score (Heusel et al., 2017) compared to Vanilla-DDPM when evaluated on CIFAR-10. We attribute this to the significant distribution discrepancy between the training set of Stable Diffusion 1.4 (Rombach et al., 2022) and CIFAR-10. Nonetheless, CIFAKE achieves higher classification accuracy than Vanilla-DDPM, highlighting the benefits of substantial external knowledge. However, by employing more advanced training and sampling

methods, such as EDM, we may generate images that are both more similar in distribution and diverse, leading to even better classification results than CIFAKE, though without external knowledge.

Table 1: Comparison of classification accuracy with different generating methods. The real-to-syn ratio is fixed at 1:1. FID is calculated w.r.t. the CIFAR-10 training set.

| Training Dataset | | Data Amount | | External? | Quality | | Top-1 Acc |
|---|---|---|---|---|---|---|---|
| Real | Syn | Real | Syn | | FID | IS | |
| | − | | 0 | ✗ | − | − | 92.48 |
| CIFAR-10 | Vanilla-DDPM | 50k | 50k | ✗ | 15.51 | 4.69 | 93.08 (+0.60) |
| | CIFAKE | | 50k | ✓ | 27.15 | 6.14 | 93.98 (+1.50) |
| | EDM | | 50k | ✗ | 8.33 | 6.23 | **94.83** (+2.35) |

**Summary**  Pre-trained models with external knowledge typically generate images with high recognizability and diversity (reflected in the relatively high IS score (Salimans et al., 2016) of CIFAKE). However, they may exhibit a greater distribution shift from the real dataset. On the other hand, generative models trained solely on internal data may show smaller distribution differences but could be limited by the amount of available data, resulting in lower-quality images (see Figure 14 (a) and (b) in Appendix for qualitative results). Even with sufficient data and cutting-edge generation methods like EDM (Karras et al., 2022), model parameters often need to be re-adjusted for different datasets based on factors such as dataset size and image resolution. Therefore, given the convenience of using pre-trained models, we utilize them in the following study.

### 3.3 GENERATIVE MODELS TRAINED ON EXTERNAL DATA FOR IMAGENET CLASSIFICATION

In the following sections, we systematically explore the trend of how synthetic data affects classification performance. To provide a more comprehensive analysis, we conduct experiments in two settings: supervised image classification and zero-shot image classification.

#### 3.3.1 SUPERVISED IMAGE CLASSIFICATION

**Experiment Setup**  The following experiments are conducted using subsets of ImageNet. Specifically, we first focus on 10 random selected classes, referred to as **ImageNet-10** (details can be found in Appendix A), to draw our primary conclusions. We then extend these observations to a larger dataset, **ImageNet-100**, as introduced in (Tian et al., 2020). We use a ResNet-50 model as the backbone architecture for all experiments in this part. We focus on external generative data augmentation in this section.

To generate synthetic images on ImageNet-10, we employ two different generation protocols with varying sample qualities. The first protocol uses Stable Diffusion 2 (Ramesh et al., 2022) with a straightforward class-conditioned prompt of the form $p_c =$ "High-quality photo of a $c$", where $c$ represents the class name. The second protocol uses Stable Diffusion 3 (Esser et al., 2024) with diverse captions generated according to the method described in (Tian et al., 2024). The caption templates include $c \rightarrow caption$, $c, bg \rightarrow caption$, and $c, rel \rightarrow caption$. We refer readers to the original paper for further details on this method. Each caption generates five images, and we employ the CLIP-Filter strategy (He et al., 2022) to exclude the bottom 20% of images based on CLIP zero-shot classification confidence, retaining only the high-quality images. For the ImageNet-100 setting, we only use the second protocol to generate synthetic data. We denote the three generated dataset as **ImageNet-10-SD2**, **ImageNet-10-SD3** and **ImageNet-100-SD3**, respctively.

Then, we consider the following scenarios for mixed training with each synthetic dataset: 1). Fixing the number of real samples at 65, 260, and 1,300 **per class** and changing the proportion of synthetic samples from $1 : 0.1$ to $1 : 100$. 2). Fixing the number of synthetic samples at 1,300 per class and varying the proportion of real samples. We conduct extensive experiments using different real and synthetic datasets, baseline sample numbers, and mixing ratios. The complete experimental results are in Appendix C.3.

**Empirical equivalence**  Similar to the experiments on internal generative data augmentation, we analyze the relationship between accuracy and the amounts of additional real and synthetic data at

various fixed values of $n_{\text{base}}$. A contour map is provided in Figure 2 (b) for $n_{\text{base}} = 26,000$. We then fitted an empirical function, as shown in Eq. 2, to roughly assess the effectiveness of synthetic data under the external setting.

**Main Results**    *Synthetic data is much less sample-efficient than real data.*

We compare the evaluation accuracy after training separately on ImageNet-100 and ImageNet-100-SD3. With 1,300 training images per class, the former achieves an accuracy of 85.41%, while the latter only reaches 50.93%. Further observation of Figure 4 reveals that when we fix 1,300 synthetic images per class and add only 0.01 times the synthetic amount (only **13** real images per class), the accuracy significantly improves by **7.95%** while adding 0.1 times the amount of synthetic data results in a remarkable enhancement of **22.62%**. This indicates that real data is much more sample-efficient than synthetic data.

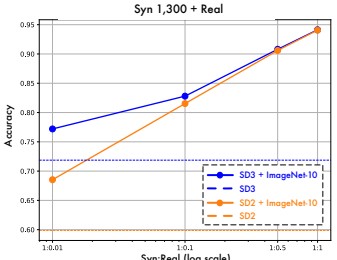
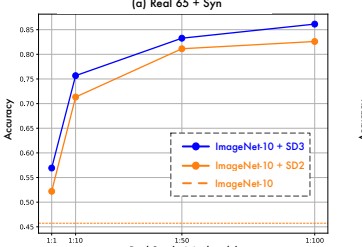
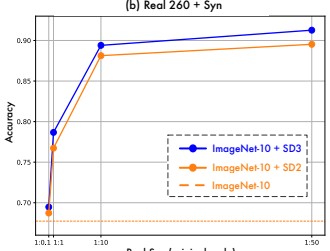

Figure 4: **Accuracy comparison** based on different synthetic data quality and real data ratio.

Figure 5: **Accuracy relative to the synthetic data ratio** at fixed real data quantities of 650 and 2,600. We use the original, unscaled proportions to illustrate the saturating effect of synthetic data.

*Integrating synthetic data greatly enhances classification performance when real data is scarce, but the benefit decreases as real data becomes more plentiful.*

We explore how the accuracy curve changes when we use different base amounts of real data and add synthetic data to it, and the results are shown in Figure 7. We observe that as the amount of real data increases, the slope of the accuracy curve with added synthetic data diminishes rapidly. This trend is consistent across ImageNet-10-SD2, ImageNet-10-SD3, and ImageNet-100-SD3.

When the amount of real data is large and classifier accuracy is already high, comparing absolute improvements may not be sufficiently rigorous. Therefore, we additionally compare the ratio of accuracy improvements when augmenting the existing real dataset with an equal proportion of synthetic data and real data. We refer to this improvement ratio as IR, which can be mathematically expressed as:

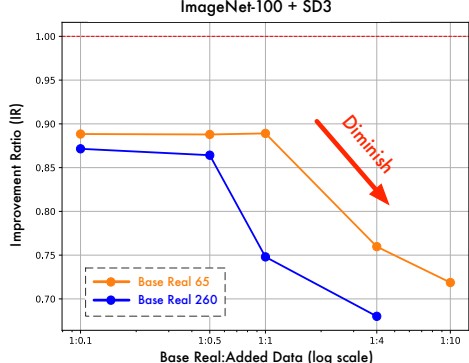

Figure 6: **The improvement ratio** IR with respect to added data ratio $r_+$. The values of $n_{\text{base}}$ are fixed at 65 and 260 per class, respectively.

$$\text{IR}(n_{\text{base}}, r_+) = \frac{\Delta\text{Acc}_{\text{syn}}(n_{\text{base}}, r_+)}{\Delta\text{Acc}_{\text{real}}(n_{\text{base}}, r_+)}. \tag{3}$$

where $\Delta\text{Acc}_{\text{syn}}$ and $\Delta\text{Acc}_{\text{real}}$ represent the accuracy improvement from adding synthetic and real data to $\mathbf{S}_{\text{base}}$, respectively, and $r_+$ denotes the ratio of the added data (real or synthetic) to $n_{\text{base}}$.

This experiment is conducted in the ImageNet-100 setting. As shown in Figure 6, with a larger baseline amount of real data, IR further decreases (the blue line is lower than the yellow line), indicating the reliability of the conclusion. We attribute this to the fact that with more real data, the model already acquires sufficient knowledge from it. Moreover, since synthetic data inherently lacks diversity and has domain gaps, its contribution to performance improvement becomes more limited.

*The performance improvement brought by synthetic data quickly diminishes as its amount increases.*

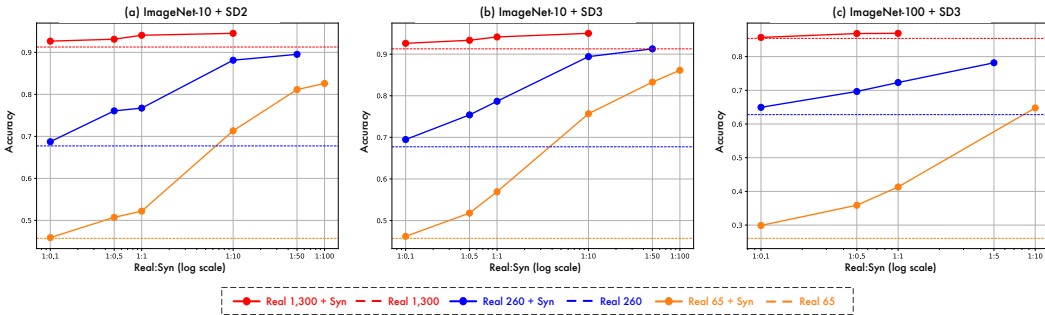

Figure 7: **Accuracy curves w.r.t. three different quantities of real images** on ImageNet-10 and ImageNet-100. Synthetic images at varying ratios are added to the training set.

In Figure 5, we plot how the accuracy changes with the synthetic data ratio while keeping the real data fixed at 65 and 260 images per class, respectively. We observe that as the amount of synthetic data increases, the improvement in classification accuracy quickly decreases. Specifically, in the case of **Real260+Syn**, adding the synthetic data from ImageNet-10-SD3 by 10 times the amount of real data raises the accuracy from 67.83% to 89.40%, a total improvement of **21.67%**. However, further adding 40 times the amount of real data only results in an additional **1.83%** increase in accuracy.

Here, we also demonstrate the saturating effect of increasing synthetic data in terms of the improvement ratio as mentioned above. As illustrated in Figure 6, both the blue and yellow lines show a decreasing trend as more data is added. We believe this is due to insufficient diversity in the synthetic data. When there is already a large amount of synthetic data, the high similarity within its internal distribution leads to the synthetic data no longer offering additional information to the classifier, resulting in only a marginal improvement in classification performance.

*The quality of synthetic data matters more when there is less amount of real data.*

As outlined in the experiment setup, we use different generation protocols to create two synthetic datasets with varying data quality on ImageNet-10. We present the details of the datasets in Table 2 and the visualization in Figure 9. It is clear that the synthetic dataset generated by SD3 exhibits superior quality. We fix the number of synthetic images per class at 1,300 and incrementally add real images to the training set at different scales. As illustrated in Figure 4, when only synthetic data is used, the model trained on the SD3-generated dataset achieves an accuracy of 71.87%, which is **12.00%** higher. However, as more real data is incorporated into the training set, the accuracy gap between the synthetic datasets gradually closes. When the amount of real data matches the synthetic data, the difference narrows to just **0.06%**, demonstrating that the quality of synthetic data is more critical when the quantity of real data is limited.

### 3.3.2 ZERO-SHOT IMAGE CLASSIFICATION

**Experiment Setup**   In the zero-shot setting, we split ImageNet-10 and ImageNet-100 into two subsets, each containing 5 and 50 categories, respectively. See appendix B for a detailed description of the settings. We will use ImageNet-100 as an example to introduce the experiment setting, with ImageNet-10 following a similar approach. The first subset of the training set is used as the real training data, while the second subset of the validation set is reserved for testing. This ensures that the model is never exposed to the real data from the categories in the validation set. To incorporate synthetic data, we apply the same split on ImageNet-100-SD3 and only retain the second subset. This is equivalent to leveraging the generative model to produce data for the test categories. With SD3-generated synthetic images, we can align all 100 classes during training. We use a pre-trained and

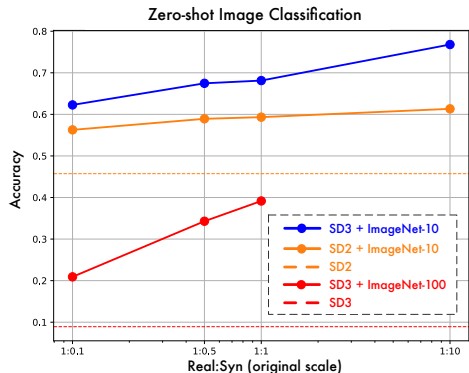

Figure 8: **Zero-shot classification accuracy** in terms of the synthetic data ratio on ImageNet-10 and ImageNet-100. The size of real data is fixed at 1,300.

frozen BERT (Devlin et al., 2019) as the text encoder and train a ResNet-50 model as the image encoder from scratch. Text and image features are projected onto a joint embedding space with a dimension of 512. The training goal is to maximize the cosine similarity between the text and image embeddings of the same categories. In the evaluation period, a test image is classified into the category with the highest similarity score. We keep the number of real images fixed at 1,300 per class and vary the proportion of synthetic data. The full experiment results are in Appendix C.4.

**Main Results** The results are shown in Figure 8. We find that adding SD3-generated images for the test categories, with just 0.1 times the amount of real data, improves accuracy by **17.54%** on ImageNet-10 and **11.99%** on ImageNet-100. Moreover, the further improvement is still notable when the synthetic data ratio reaches 1:1, demonstrating the significant potential of synthetic data in zero-shot classification. We also observe that the quality of synthetic data plays a more crucial role in the zero-shot setting, as SD3-generated images achieve **15.47%** higher accuracy at the ratio of 1:10. Our findings align with what has been observed when the amount of real data is minimal in the supervised setting. We suggest that, since zero-shot classification lacks any real data from the test categories, it can be considered a natural extension of supervised learning with diminishing real data, which explains why these results are logical and expected.

Table 2: Detailed configurations of the synthetic datasets used for image classification on ImageNet-10. Data quality is measured by FID (vs. ImageNet-10 training set) and Inception Score (IS).

| Generative Model | Data Amount | CFG Scale | Prompt | # Classes | FID | IS |
|---|---|---|---|---|---|---|
| Stable Diffusion 2 | 130k | 7.5 | "High-quality photo of a $c$." | 10 | 40.54 | 1.37 |
| Stable Diffusion 3 | 130k | 2.0 | Generated captions (Tian et al., 2024) | 10 | 24.79 | 8.30 |

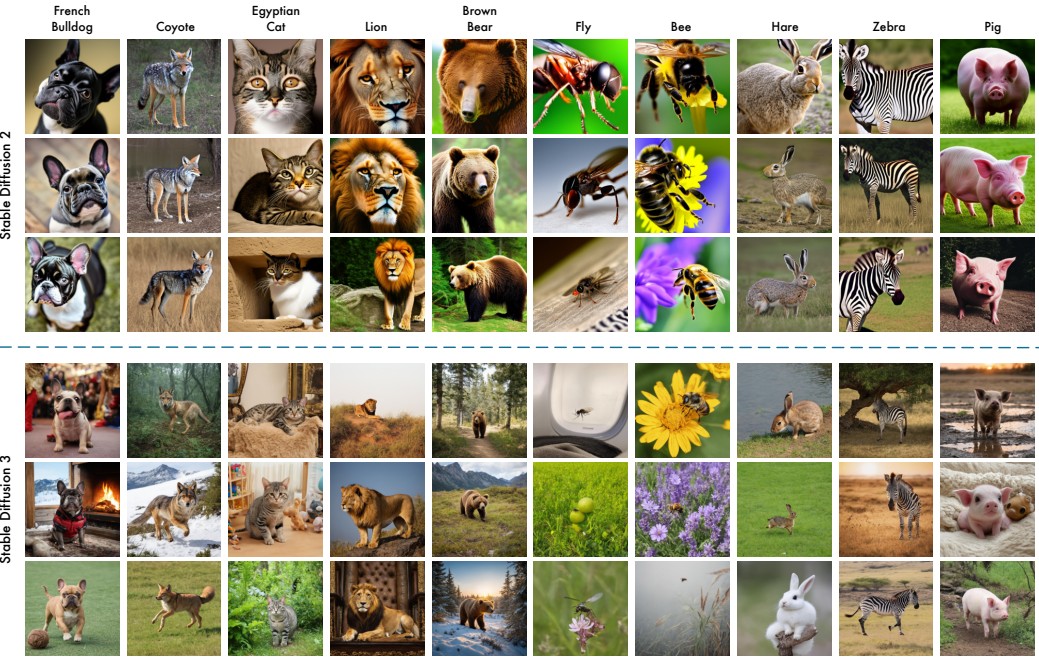

Figure 9: Visualizations of the SD2-generated and SD3-generated synthetic dataset for ImageNet-10. SD2-generated images are often object-centric, focusing predominantly on the object's face. The backgrounds, shapes, and poses are usually uniform. In contrast, SD3-generated images present a more complete view of the objects, with diverse backgrounds and varied poses.

## 4 RELATED WORK

**Synthetic Data Augmentation in Computer Vision** The usage of synthetic data augmentation in image classification has gained significant attention because of its potential to generate large amounts

of data with minimal manual effort. Synthetic images have proven effective across various computer vision tasks, including semantic image segmentation (Chen et al., 2019; Tritrong et al., 2021), object detection (Nowruzi et al., 2019; Fabbri et al., 2021; Zhang et al., 2022; Ge et al., 2022), human motion understanding (Guo et al., 2022; Varol et al., 2017), and 3D reconstruction (Xu et al., 2024; Zhang et al., 2023b; Wu* et al., 2022). Adversarial data augmentation (Xie et al., 2020) has shown to improve image recognition. They provide diverse and comprehensive training data to improve model generalization. Early methods primarily rely on simulation pipelines using graphics engines or 2D renderings to generate synthetic data (Dosovitskiy et al., 2015; 2017). However, these approaches often encounter high computational costs, which can limit their scalability.

Graphics simulations (Beery et al., 2020) have been used to perform synthetic data augmentation for image recognition. More recent approaches have explored the use of generative models to generate synthetic data for image classification (Sarıyıldız et al., 2023; Zhou et al., 2023; Bansal & Grover, 2023; Hennicke et al., 2024; Jung et al., 2024). Text-to-image diffusion models, in particular, have gained prominence as these models can generate high-quality, large-scale curated datasets with just a few textual descriptions. For instance, He et al. (2022) has found that synthetic data generated by GLIDE (Nichol et al., 2021) can readily benefit image classification in data-scarce settings. Trabucco et al. (2024) proposes a data augmentation method that uses pre-trained text-to-image diffusion models to enhance semantic diversity in images, leading to improved accuracy in few-shot image classification tasks. Azizi et al. (2023) has demonstrated that fine-tuning Imagen (Saharia et al., 2022) using a target dataset can improve classification accuracy. Fan et al. (2024) studies the scaling laws of synthetic images generated by text-to-image diffusion models to train image classifiers.

Unlike their approach, we answer a more fundamental and overarching question: When we have a certain amount of real data for classification, how **quantitatively** can synthetic data augmentation help with image classification? How does the role of synthetic data vary under different scales of real data?

**Text-to-Image Diffusion Models**   Diffusion models (Ho et al., 2020b; Song et al., 2020a;b) have emerged as powerful generative models capable of producing high-quality, photo-realistic images. Comparing with traditional generative adversarial networks (Goodfellow et al., 2014; Tu, 2007), diffusion models offer comparable or even superior image quality while also providing greater training stability. Specifically, text-to-image (T2I) diffusion models enable flexible language prompts to generate diverse and customized images. Imagen (Saharia et al., 2022), Stable Diffusion (Rombach et al., 2022), DALL-E (Ramesh et al., 2021), Muse (Chang et al., 2023), and GLIDE (Nichol et al., 2021) are notable T2I models. Additionally, ControlNet (Zhang et al., 2023a), T2-Adapter (Mou et al., 2023), UniControl (Qin et al., 2023), and OmniControlNet (Wang et al., 2024) demonstrate excellent capabilities in image-conditioned text-to-image tasks.

In this work, we focus on Stable Diffusion, a latent diffusion model (LDM) that performs the diffusion process within the latent space of the Variational AutoEncoder (Kingma, 2013; Van Den Oord et al., 2017). This approach significantly reduces computational demands compared to pixel-based models while achieving superior visual fidelity and performance across various tasks.

## 5 CONCLUSION

In this study, we present Mousterian, an empirical study that systematically explores how synthetic data can enhance classification models, starting from fundamental classification tasks, and identify scenarios where synthetic data proves particularly effective. Through a series of experiments, we demonstrate the efficacy of direct mixed training and reveal that generative models have the potential to improve classification performance, regardless of the involvement of external datasets. Notably, we observe a significant utility of generated data when the amount of real data is limited, alongside a saturation trend in performance improvement as the data volume increases. Additionally, we provide an empirical functional relationship between accuracy and the amount of real and synthetic data added, aiming to offer researchers an intuitive understanding of this relationship. We hope that our findings will provide valuable insights for future research on synthetic data in computer vision.

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

# A    IMAGENET-10

ImageNet-10 comprises 10 classes randomly selected from the original ImageNet-1k (Deng et al., 2009a). Each class contains roughly 1,300 images. The class labels are French bulldog, coyote, Egyptian cat, lion, brown bear, fly, bee, hare, zebra, and pig. The visualization of ImageNet-10 is provided in Figure 10.

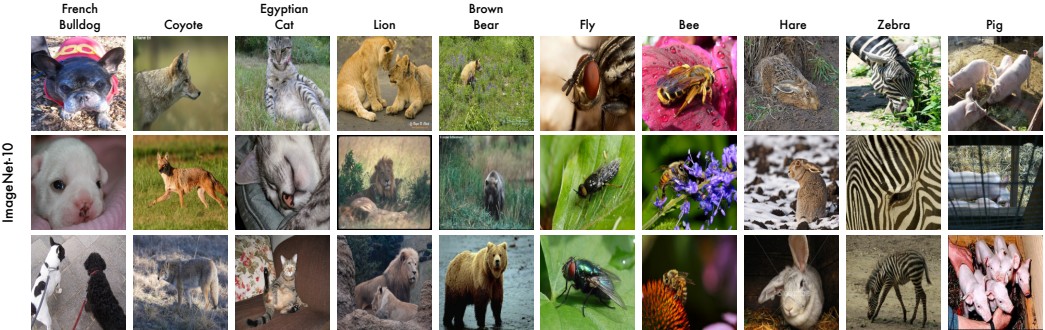

Figure 10: Visualizations of the ImageNet-10. Ten classes are randomly selected from the original ImageNet-1k.

# B    PRELIMINARIES

**Zero-shot image classification**    Zero-shot image classification can be formalized as follows: Let $\mathcal{Y}_{\text{train}}$ be the set of training categories, the training set $\mathcal{D}_{\text{train}}$ consists of samples $\{(x_i, y_i)\}_{i=1}^N$, with $x_i$ representing an image and $y_i \in \mathcal{Y}_{\text{train}}$ its class label. The validation set $\mathcal{D}_{\text{test}}$ includes samples $\{(x_j, y_j)\}_{j=1}^M$, where $x_j$ is a test image, but its class label $y_j \in \mathcal{Y}_{\text{test}}$ is not part of the training data, i.e., $\mathcal{Y}_{\text{train}} \cap \mathcal{Y}_{\text{test}} = \emptyset$. During training, the class label $y$ is first mapped to a text description $M(y)$, where $M \in \mathcal{M}$ is a natural language template. Then, a text encoder $T$ converts $M(y)$ into a feature vector, which is subsequently projected onto the joint embedding space using a linear layer, resulting in a text embedding $\text{Emb}_{\text{text}}(y)$. For images, a visual encoder processes the image $x$ to produce its feature representation $I(x)$. This feature is also projected onto the same space as the text embedding. The training goal is to maximize the cosine similarity between $\text{Emb}_{\text{text}}(y)$ and $\text{Emb}_{\text{image}}(x)$. In the testing period, given a test image $x_{\text{test}}$, it is classified into the category $\hat{y}_{\text{test}}$ with the highest similarity score. Although the categories of the test images are not seen during training, this approach enables classification by leveraging their semantic relationships with seen classes.

In our case, by leveraging the powerful capabilities of a generative model, we can create a synthetic dataset $\mathcal{D}_{\text{syn}}$ composed solely of images of $\mathcal{Y}_{\text{test}}$ and mix it with the original training set $\mathcal{D}_{\text{real}}$, resulting in a combined training set $\mathcal{D}_{\text{mixed}} = \mathcal{D}_{\text{real}} \cup \mathcal{D}_{\text{syn}}$. In this way, $\mathcal{Y}_{\text{test}}$ is included in the categories of $\mathcal{D}_{\text{mixed}}$, while the classifier has only not seen the **real** data of $\mathcal{Y}_{\text{test}}$.

# C    ADDITION EXPERIMENTS

## C.1    REAL AND SYNTHETIC DATA ARE FUNDAMENTALLY DIFFERENT

In this section, we provide more detailed information about the experiment mentioned in Section 2.1. To investigate the differences in data distribution, we conduct extensive experiments across various datasets using different synthetic data generation methods. A domain classifier is trained to distinguish between input images from the real domain and the synthetic domain. Considering both global and local statistical differences, we evaluate two scenarios: one where the input images are original images and another where the input images are image patches.

The real datasets include CIFAR-10, ImageNet-10, and ImageNet-100. For CIFAR-10, we utilize datasets generated by conditional Vanilla-DDPM and EDM, as introduced in previous sections, as synthetic datasets. Given the image resolution of these datasets is $32 \times 32$, we only consider scenarios where the input images are the original images. For ImageNet-10, we employ ImageNet-10-SD2 and

ImageNet-10-SD3 as synthetic datasets, conducting experiments on both original images and image patches. For ImageNet-100, we use ImageNet-100-SD3. All synthetic datasets are the same size as the corresponding real dataset. The results are presented in Figure 11. All experiments involving the classification of image patches achieve an accuracy greater than 90%, while those involving the classification of full images achieve an accuracy exceeding 98%. These findings demonstrate a fundamental difference in the distribution between real and synthetic data.

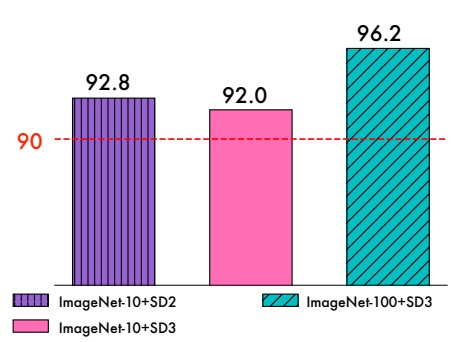
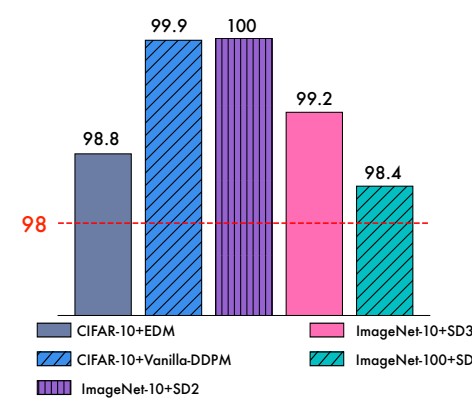

Figure 11: Visualizations of the domain classification experiments on image patches (a) and original images (b). The numbers in the figure are the accuracy (%).

### C.2 Full results on CIFAR-10

We provide the results used for fitting the equivalence contour under the internal setting in Section 3.1, as shown in Figure 3. In Section 3, all ResNet models are trained from scratch. Here, we investigate whether synthetic data still provides benefits for classifiers pre-trained on large-scale datasets. To explore this, we use a ViT-L/16 model pre-trained on ImageNet-21K. After fine-tuning this model on CIFAR-10, it achieves near-SOTA performance (**99.22%** in our tests). We also fine-tune the model on a mixed dataset of CIFAR-10 and CIFAKE synthetic datasets, which results in **99.23%**, showing virtually no improvement. We believe this is because the pre-trained model has already learned extensive knowledge from large-scale real data, and factors such as the quality and distribution differences of synthetic data limit its usefulness when fine-tuning the model.

### C.3 Full results on supervised image classification

We provide the full results of supervised image classification on different datasets, data amount, and data ratio. Results on ImageNet-10 and two synthetic datasets with different quality (ImageNet-10-SD2, ImageNet-10-SD3) are provided in Table 4. Results on ImageNet-100 (Tian et al., 2020) and the corresponding synthetic dataset (ImageNet-100-SD3) are shown in Table 5. Information on ImageNet-10 can be found in Appendix A.

Additionally, we provide a comprehensive comparison of the impact of different real data amounts, real-to-synthetic data ratios, and synthetic data quality on model classification performance for ImageNet-10, as shown in Figure 12. It can be observed that the gap between the two blue curves is much larger than the gap between the two orange curves, which is greater than that between the red curves. Thus, we can conclude that as the amount of real data increases, the impact of synthetic data quality diminishes, which aligns with the conclusions drawn in Section 3.3.1.

### C.4 Full results on zero-shot image classification

We conduct zero-shot image classification experiments on ImageNet-10 and ImageNet-100 with three synthetic datasets. The model is evaluated on the validation set of categories whose real images it does not see during training. The results are provided in Table 6.

Table 3: Experimental results on CIFAR-10 used for fitting the equivalence contour for internal generative data augmentation.

| Training Dataset | | Data Amount | | | Acc |
|---|---|---|---|---|---|
| Real | Syn | Real | Syn | Total | Top-1 |
| | | 5,000 | 0 | 0 | 67.05 |
| | | | 5,000 | 10,000 | 71.63 |
| | | 25,000 | 0 | 25,000 | 87.35 |
| | | | 10,000 | 35,000 | 88.08 |
| | | | 20,000 | 45,000 | 88.46 |
| | | | 25,000 | 50,000 | 88.41 |
| | | | 30,000 | 55,000 | 88.80 |
| | | | 40,000 | 65,650 | 88.44 |
| | | | 50,000 | 75,000 | 89.10 |
| | | | 60,000 | 85,000 | 89.19 |
| | | | 70,000 | 95,000 | 89.96 |
| CIFAR-10 | Vanilla-DDPM | | 75,000 | 100,000 | 89.15 |
| | | | 80,000 | 105,000 | 89.41 |
| | | | 90,000 | 115,000 | 90.03 |
| | | | 100,000 | 125,000 | 89.51 |
| | | 30,000 | | 30,000 | 87.65 |
| | | 32,500 | | 32,500 | 89.3 |
| | | 35,000 | | 35,000 | 89.64 |
| | | 37,500 | 0 | 37,500 | 89.75 |
| | | 40,000 | | 40,000 | 91.22 |
| | | 42,500 | | 42,500 | 91.17 |
| | | 45,000 | | 45,000 | 91.78 |
| | | 47,500 | | 47,500 | 92.53 |
| | | 50,000 | 0 | 50,000 | 92.48 |
| | | | 50,000 | 100,000 | 93.08 |
| | | | 100,000 | 150,000 | 92.64 |
| | | | 150,000 | 200,000 | 92.92 |
| | | | 200,000 | 250,000 | 93.05 |
| | | | 250,000 | 300,000 | 92.76 |

Table 4: Experimental details of supervised image classification on ImageNet-10. **Real:Syn** refers to the ratio of the quantity of real data to synthetic data used in training.

| Training Dataset | | Data Amount | | | Real:Syn | Acc |
|---|---|---|---|---|---|---|
| Real | Syn | Real | Syn | Total | | Top-1 |
| ImageNet-10 | ImageNet-10-SD2 | 650 | 0 | 650 | real only | 45.73 |
| | | | 65 | 715 | 1:0.1 | 46.20 (+0.47) |
| | | | 325 | 975 | 1:0.5 | 50.73 (+5.00) |
| | | | 650 | 1,300 | 1:1 | 52.20 (+6.47) |
| | | | 6,500 | 7,150 | 1:10 | 71.33 (+25.6) |
| | | | 32,500 | 33,150 | 1:50 | 81.13 (+35.40) |
| | | | 65,000 | 65,650 | 1:100 | **82.60 (+36.87)** |
| | | 2,600 | 0 | 2,600 | real only | 67.73 |
| | | | 260 | 2,860 | 1:0.1 | 68.73 (+1.00) |
| | | | 1,300 | 3,900 | 1:0.5 | 76.07 (+8.34) |
| | | | 2,600 | 5,200 | 1:1 | 76.73 (+9.00) |
| | | | 26,000 | 28,600 | 1:10 | 88.13 (+20.40) |
| | | | 130,000 | 132,600 | 1:50 | **89.53 (+21.80)** |
| | | 13,000 | 0 | 13,000 | real only | 91.27 |
| | | | 1,300 | 14,300 | 1:0.1 | 92.67 (+1.40) |
| | | | 6,500 | 19,500 | 1:0.5 | 93.13 (+1.86) |
| | | | 13,000 | 26,000 | 1:1 | 94.07 (+2.80) |
| | | | 130,000 | 143,000 | 1:10 | **94.53 (+3.26)** |
| | | 0 | | 13,000 | syn only | 59.87 |
| | | 130 | | 13,130 | 1:100 | 68.53 (+8.66) |
| | | 1,300 | 13,000 | 14,300 | 1:10 | 81.53 (+21.66) |
| | | 6,500 | | 19,500 | 1:2 | 90.60 (+30.73) |
| | | 13,000 | | 26,000 | 1:1 | **94.07 (+34.20)** |
| ImageNet-10 | ImageNet-10-SD3 | 650 | 0 | 650 | real only | 45.73 |
| | | | 65 | 715 | 1:0.1 | 46.20 (+0.47) |
| | | | 325 | 975 | 1:0.5 | 51.80 (+6.07) |
| | | | 650 | 1,300 | 1:1 | 56.93 (+11.20) |
| | | | 6,500 | 7,150 | 1:10 | 75.67 (+29.94) |
| | | | 32,500 | 33,150 | 1:50 | 83.27 (+37.54) |
| | | | 65,000 | 65,650 | 1:100 | **86.13 (+40.40)** |
| | | 2,600 | 0 | 2,600 | real only | 67.73 |
| | | | 260 | 2,860 | 1:0.1 | 69.47 (+1.74) |
| | | | 1,300 | 3,900 | 1:0.5 | 75.40 (+7.67) |
| | | | 2,600 | 5,200 | 1:1 | 78.67 (+10.94) |
| | | | 26,000 | 28,600 | 1:10 | 89.40 (+21.67) |
| | | | 130,000 | 132,600 | 1:50 | **91.27 (+23.54)** |
| | | 13,000 | 0 | 13,000 | real only | 91.27 |
| | | | 1,300 | 14,300 | 1:0.1 | 92.60 (+1.33) |
| | | | 6,500 | 19,500 | 1:0.5 | 93.33 (+2.06) |
| | | | 13,000 | 26,000 | 1:1 | 94.13 (+2.86) |
| | | | 130,000 | 143,000 | 1:10 | **95.00 (+3.73)** |
| | | 0 | | 13,000 | syn only | 71.87 |
| | | 130 | | 13,130 | 1:100 | 77.20 (+5.33) |
| | | 1,300 | 13,000 | 14,300 | 1:10 | 82.80 (+10.93) |
| | | 6,500 | | 19,500 | 1:2 | 90.80 (+18.93) |
| | | 13,000 | | 26,000 | 1:1 | **94.13 (+22.26)** |

Table 5: Performance comparison on ImageNet-100 (Tian et al., 2020).

| Training Dataset | | Data Amount | | | Real:Syn | Acc | |
|---|---|---|---|---|---|---|---|
| Real | Syn | Real | Syn | Total | | Top-1 | Top-5 |
| ImageNet-100 | ImageNet-100-SD3 | 6,500 | 0 | 6,500 | real only | 26.07 | 49.93 |
| | | | 650 | 7,150 | 1:0.1 | 29.89 (+3.82) | 54.29 (+4.36) |
| | | | 3,250 | 9,750 | 1:0.5 | 35.89 (+9.82) | 60.22 (+10.29) |
| | | | 6,500 | 13,000 | 1:1 | 41.31 (+15.24) | 66.05 (+16.12) |
| | | | 65,000 | 71,500 | 1:10 | **64.78** (+38.71) | **85.83** (+35.90) |
| | | 26,000 | 0 | 26,000 | real only | 62.78 | 82.90 |
| | | | 2,600 | 28,600 | 1:0.1 | 64.95 (+2.17) | 84.71 (+1.81) |
| | | | 13,000 | 39,000 | 1:0.5 | 69.65 (+6.87) | 88.11 (+5.21) |
| | | | 26,000 | 52,000 | 1:1 | 72.31 (+9.53) | 89.81 (+6.91) |
| | | | 130,000 | 156,000 | 1:5 | **78.17** (+15.39) | **93.49** (+10.59) |
| | | 130,000 | 0 | 130,000 | real only | 85.41 | 96.39 |
| | | | 13,000 | 143,000 | 1:0.1 | 85.69 (+0.28) | 96.63 (+0.24) |
| | | | 65,000 | 195,000 | 1:0.5 | 86.85 (+1.44) | 97.10 (+0.71) |
| | | | 130,000 | 260,000 | 1:1 | **86.91** (+1.50) | **97.29** (+0.90) |
| | | 0 | | 130,000 | syn only | 50.93 | 76.63 |
| | | 1,300 | | 131,300 | 1:100 | 58.88 (+7.95) | 82.53 (+5.90) |
| | | 13,000 | 130,000 | 143,000 | 1:10 | 73.55 (+22.62) | 91.07 (+14.44) |
| | | 65,000 | | 195,000 | 1:2 | 83.49 (+32.56) | 95.71 (+19.08) |
| | | 130,000 | | 260,000 | 1:1 | **86.91** (+35.98) | **97.29** (+20.66) |

Table 6: Full experiments results of zero-shot image classification on ImageNet-10 and ImageNet-100 with corresponding synthetic datasets.

| Training Dataset | | Data Amount | | | Real:Syn | Acc |
|---|---|---|---|---|---|---|
| Real | Syn | Real | Syn | Total | | Top-1 |
| ImageNet-10 | ImageNet-10-SD2 | 6,500 | 0 | 6,500 | real only | 45.73 |
| | | | 650 | 7,150 | 1:0.1 | 56.27 (+10.54) |
| | | | 3,250 | 9,750 | 1:0.5 | 58.93 (+13.20) |
| | | | 6,500 | 13,000 | 1:1 | 59.33 (+13.60) |
| | | | 65,000 | 71,500 | 1:10 | **61.33** (+16.60) |
| ImageNet-10 | ImageNet-10-SD3 | 6,500 | 0 | 6,500 | real only | 45.73 |
| | | | 650 | 7,150 | 1:0.1 | 62.27 (+16.54) |
| | | | 3,250 | 9,750 | 1:0.5 | 67.47 (+21.74) |
| | | | 6,500 | 13,000 | 1:1 | 68.13 (+22.40) |
| | | | 65,000 | 71,500 | 1:10 | **76.80** (+31.07) |
| ImageNet-100 | ImageNet-100-SD3 | 130,000 | 0 | 130,000 | real only | 8.92 |
| | | | 13,000 | 143,000 | 1:0.1 | 20.91 (+11.99) |
| | | | 65,000 | 195,000 | 1:0.5 | 34.29 (+25.37) |
| | | | 130,000 | 260,000 | 1:1 | **39.16** (+30.24) |

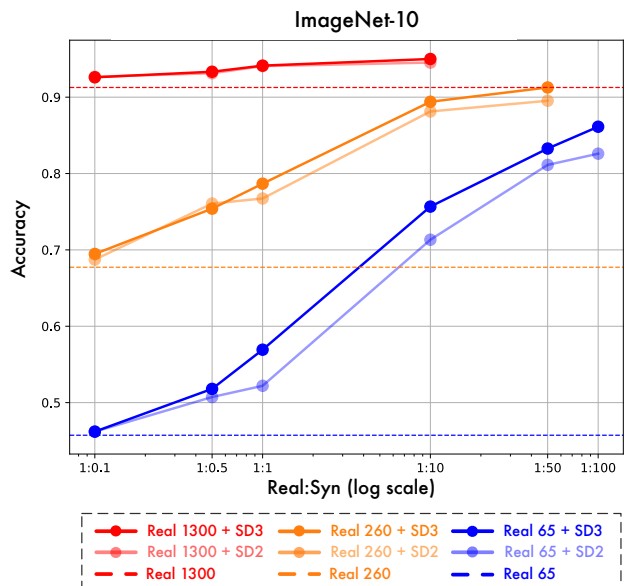

Figure 12: Classification accuracy relative to the ratio of synthetic data on ImageNet-10. Different colors represent different baseline amounts of real data, while different levels of transparency indicate different synthetic data quality.

## D IMPLEMENTATION DETAILS

Both supervised and zero-shot image classification experiments are performed on CIFAR-10, ImageNet-10, ImageNet-100, and various corresponding synthetic datasets. For ImageNet-10 and ImageNet-100 experiments, we fix the number of total iterations instead of total epochs. However, since our mixed dataset size varies significantly, ranging from 650 to 260,000, we group the total number of iterations into different levels based on the data size, and the details are provided in Table 7. All images are resized to 224 × 224 for input. For data augmentation, we apply random cropping, resizing, and random horizontal flipping.

For the CIFAR-10 experiments, we use ResNet-110 as introduced in (He et al., 2016) for classification, as it is well-suited for the smaller image sizes of CIFAR-10. In the experiments on CIFAR-Half and the full CIFAR-10, we fix the number of epochs at 160, using a multistep scheduler to decay the learning rate by a factor of 0.1 at epochs 80 and 120. For CIFAR-Small, we fix the number of epochs at 320, also using a multistep scheduler, with the learning rate decaying by a factor of 0.1 at epochs 160 and 240. For data augmentation, we apply random cropping and random horizontal flipping.

For the ViT-L/16 (Kolesnikov et al., 2021) fine-tuning experiment, we fix the total iterations at 10,000 steps. The 32 × 32 CIFAR-10 images are resized to 224 × 224 for input.

For all settings mentioned above, we run **3** trials for each experiment and report the average result. Details on other hyper-parameters are provided in Table 8.

In our mixed training approach, we utilize PyTorch's (Paszke et al., 2019) `ConcatDataset` method to combine real and synthetic data. The `RandomSampler` of PyTorch randomly shuffles the combined dataset at the beginning of each epoch.

When calculating the Frechet Inception Distance (FID) score between synthetic datasets and real datasets, we employ the official implementation of FID to PyTorch (Seitzer, 2020).

Table 7: Iterations of the ImageNet-10 and ImageNet-100 experiments with respect to data amount.

| Data amount | [650, 1.3k) | [1.3k, 2.6k) | [2.6k, 13k) | [13k, 260k] |
|---|---|---|---|---|
| Iterations | 10k | 30k | 60k | 120k |

Table 8: Hyperparameters used to train ResNet-50, ResNet-110, and ViT-L/16.

| Hyper-parameter | ResNet-50 | ResNet-110 | ViT-L/16 |
|---|---|---|---|
| Batch size | 192 | 128 | 512 |
| Base lr | 0.1 | 0.1 | 0.03 |
| Decay method | cosine | multistep | cosine |
| Optimizer | SGD | SGD | SGD |
| Momentum | 0.9 | 0.9 | 0.9 |
| Weight decay | 1e-4 | 1e-4 | 0 |
| Warmup iterations | 10% | no warmup | 5% |

## E    ADDITION VISUALIZATIONS

### E.1    VISUALIZATION: SYNTHETIC DATA FOR IMAGENET-100

We visualize the synthetic dataset for ImageNet-100, generated by Stable Diffusion 3. Ten classes are randomly selected from the entire dataset. All example images are randomly sampled from their respective classes without any manual curation. The visualizations are presented in Figure 13.

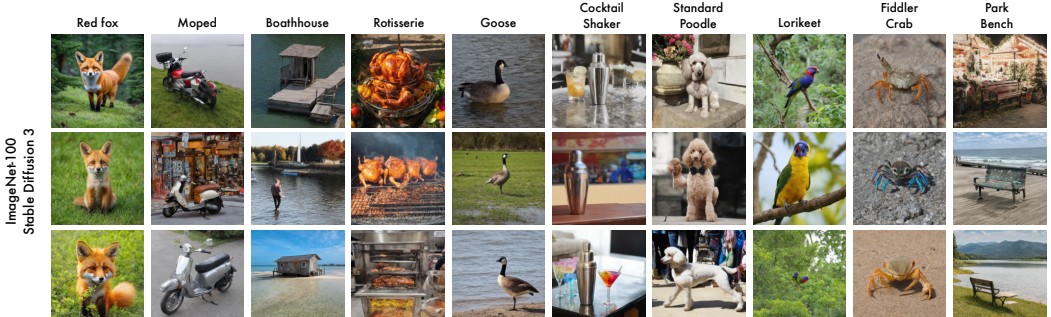

Figure 13: Visualizations of the SD3-generated synthetic dataset for ImageNet-100.

### E.2    VISUALIZATION: SYNTHETIC DATA FOR CIFAR-10

We visualize some example images of the synthetic datasets for CIFAR-10, including three datasets generated by Vanilla-DDPM (Ho et al., 2020b), which is trained on CIFAR-Small, CIFAR-Half, and the full CIFAR-10, respectively, a dataset sampled using EDM (Karras et al., 2022), and the synthetic dataset from CIFAKE (Ho et al., 2020b). The visualizations are shown in Figure 14.

Qualitatively, both the EDM-generated dataset and CIFAKE have relatively high recognizability. However, CIFAKE images exhibit domain shifts; for example, the ship in the third row in Figure 14 (d) is generated as an interior scene rather than its external form. The Vanilla-DDPM model trained on the full CIFAR-10 produces some distorted images, such as the frog in the first row and the cat in the third row in Figure 14 (c), which explains its significantly lower IS score. The image quality further declines for the Vanilla-DDPM models trained on CIFAR-Half and CIFAR-Small. Certain images, such as those of cats and dogs, become almost unrecognizable to the human eye.

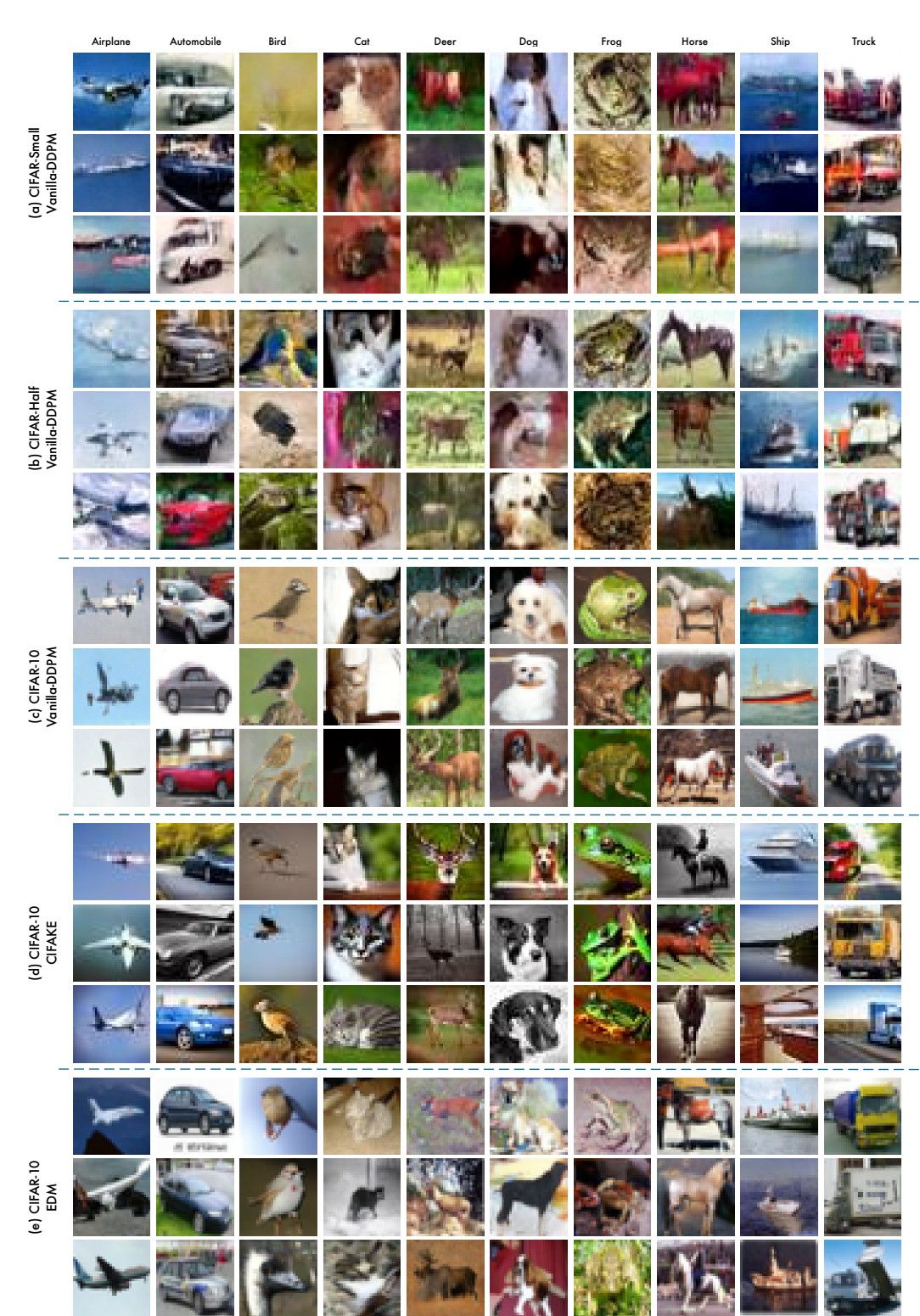

Figure 14: Visualizations of the synthetic datasets for CIFAR-10, CIFAR-Half, and CIFAR-Small.

