# OpenReview forum: "Mousterian: exploring the equivalence of generative and real data augmentation in classification"
_ICLR.cc/2025/Conference — ICLR 2025 Conference Withdrawn Submission_

### Official Review · Reviewer_TBMi · 2024-10-15

**Soundness:** 2
**Presentation:** 3
**Contribution:** 2
**Rating:** 3
**Confidence:** 4

**Summary:**

This paper explores the effectiveness of generative data augmentation for image classification, focusing on models trained on internal datasets as well as those trained on large-scale external datasets. The authors conduct experiments on CIFAR-10 and ImageNet and find that while generative data augmentation can enhance classification performance, it generally requires significantly more synthetic data to achieve similar effects as real data augmentation. The study provides practical guidelines for quantifying the ratio between generative and real data needed to achieve comparable improvements. The results suggest that high-quality synthetic data can significantly improve performance when real data is limited, though it remains less efficient compared to real data.

**Strengths:**

1. The paper presents a novel investigation into the role of generative data augmentation for image classification, focusing on both internal (training set only) and external (pre-trained on large datasets) generative models. It uniquely addresses the equivalence between synthetic and real data augmentation, providing empirical quantification to match their effects.

2. The paper is well-constructed, featuring rigorous experimentation on widely recognized datasets like CIFAR-10 and ImageNet. The thorough comparison of real vs. synthetic data, using multiple generative models, ensures that the findings are robust and generalizable.

3. The findings offer practical guidelines for leveraging synthetic data when real data is scarce, providing valuable insights into the trade-offs between real and synthetic data. The results have broad implications for data augmentation in machine learning, especially in contexts where data collection is challenging.

**Weaknesses:**

1. The paper's empirical analysis primarily focuses on CIFAR-10 and ImageNet, which limits the generalizability of the findings to more complex or diverse datasets. It would be beneficial to include experiments on additional datasets with varying complexity to validate the proposed approach across different domains.

2. The equivalence formulation between real and synthetic data is presented without sufficient theoretical backing. While the empirical results are insightful, the lack of a theoretical framework makes it difficult to generalize the equivalence relationship beyond the specific datasets and models used in the study.

3. The paper does not deeply address the quality of synthetic data and how it impacts classification performance beyond a certain point. For instance, Figures 1(b) and 1(c) highlight the impact of synthetic data on performance, but a more detailed analysis of the characteristics of synthetic data (e.g., diversity, fidelity) is needed to explain the diminishing returns observed in these figures.

4. The experiments on internal and external generative models are treated as largely separate investigations, without adequately exploring potential synergies. Figure 2 shows equivalence curves for internal and external settings separately, but it would be valuable to examine whether combining internal and external generative models could lead to better data augmentation strategies, especially in settings with limited labeled data.

5. The discussion on the trade-offs between real and synthetic data augmentation could be expanded to consider the computational cost of generating synthetic data. Practical considerations, such as the time and resources required to generate large synthetic datasets compared to collecting real data, are not sufficiently addressed. Figure 5, which illustrates the diminishing returns of synthetic data augmentation, could benefit from a discussion on the associated computational costs to assess the feasibility of the proposed approach in real-world scenarios.

6. The study mainly uses accuracy as the evaluation metric, which may not fully capture the nuances of model performance. For example, Figures 4 and 7 show accuracy improvements with different synthetic data ratios, but incorporating additional metrics, such as robustness to adversarial attacks or generalization to out-of-distribution samples, could provide a more comprehensive evaluation of the benefits and limitations of synthetic data augmentation.

**Questions:**

1. Could the authors provide more theoretical insights or justification for the empirical equivalence between synthetic and real data augmentation? How generalizable is this relationship across different types of datasets and generative models?

2. Have the authors considered combining internal and external generative models to leverage the benefits of both? If so, how would such a hybrid approach affect classification performance, particularly in cases with limited labeled data?

3. Could the authors provide an analysis comparing the computational cost of generating synthetic data to the benefits it provides in terms of performance gains? This would help in assessing the practical feasibility of using generative models for data augmentation in real-world applications.

4. Would the authors consider including other evaluation metrics, such as robustness to adversarial attacks or generalization to out-of-distribution samples, to provide a more holistic assessment of the classifier's performance when trained with synthetic data?

5. Is there an optimal ratio of synthetic to real data that consistently yields the best performance, as observed in Figure 5? If so, how is this ratio affected by the quality of the synthetic data or the complexity of the underlying dataset?

---

> ### Author Response · Authors · 2024-11-14
>
> Many thanks to Reviewer TBMi for providing detailed suggestions.
>
> > W1: "include experiments on additional datasets with varying complexity to validate the proposed approach across different domains."
>
> We consider that ImageNet and CIFAR-10 are widely recognized as representative datasets in image classification, making conclusions drawn from them broadly applicable and indicative of general trends.
>
> > W2 & Q1: "theoretical insights or justification for the empirical equivalence between synthetic and real data augmentation, and the  generalization of this relationship across different types of datasets and generative models."
>
> Our formula is not derived theoretically but rather an empirical fit to the data points. While it may not be highly precise, it effectively reflects our conclusions by illustrating the quantitative equivalence between real and generative data augmentation.
>
> For the generalization of this relationship, although the relationship is derived on CIFAR-10+Vanilla-DDPM and ImageNet-100+Stable Diffusion 3 settings, we assume this form of relationship will generalize to other **image classification scenarios**. Note that the coefficients in the formula may change, but we expect the form to hold. We leave validating on more image classification datasets for future work.
>
> > W3:"a more detailed analysis of the characteristics of synthetic data (e.g., diversity, fidelity) is needed to explain the diminishing returns observed in these figures"
>
>
> Previous work [1] has explored this issue and found that both recognizability and diversity are essential for synthetic data to enhance image classification effectively. Our focus, however, is not on improving synthetic data generation but on understanding the general trend of its effectiveness, which does not change with varing data quality.
>
> > W4 & Q2: "examine whether combining internal and external generative models could lead to better data augmentation strategies,"
>
> This is a valuable suggestion which can be explored in future work. However, in this paper, we mainly focus on the internal and external settings, which are more common in most synthetic data application scenarios. We will explore this in the future.
>
> > W5 & Q3: "consider the computational cost of generating synthetic data. discuss on the associated computational costs to assess the feasibility of the proposed approach in real-world scenarios."
>
> We have conducted experiments to record the computational cost of generating synthetic data, as shown below:
>
> 1. CIFAR-10 Experiments (A4500 GPU)
> - **EDM**: 18 sampling steps, batch size of 100
>   - Memory usage: 4000MB
>   - Time per image: ~0.06 seconds
>
> - **Vanilla-DDPM**: 500 sampling steps, batch size of 100
>   - Memory usage: 3300MB
>   - Time per image: ~0.95 seconds
>
> We used 4 A4500 GPU, and it took 16 hours to generate Vanilla-DDPM data at 5 times the scale of CIFAR-10.
>
> 2. ImageNet Experiments (A6000 GPU)
> - **Stable Diffusion 2**: 50 sampling steps
>   - Memory usage: ~3900MB
>   - Time per image: ~1 second
>
> - **Stable Diffusion 3**: 28 sampling steps
>   - Memory usage: ~21800MB
>   - Time per image: ~7 seconds
>
> We used 4 A6000 GPU, and it took 2~3 days to generate SD-3 synthetic data at the same scale as ImageNet-100.
>
> Both computational costs are affordable in our settings.
> > W6 & Q4: "incorporating additional metrics, such as robustness to adversarial attacks or generalization to out-of-distribution samples."
>
> This is a helpful suggestion. Previous works [1], [2] have explored this issue. Our focus, however, is on understanding **the effectiveness of generative data augmentation for image classification within a given dataset**. Thus, accuracy remains our primary concern, while robustness and adversarial attack resistance are viewed as potential extensions to the core problem.
>
> > Q5: "Is there an optimal ratio of synthetic to real data that consistently yields the best performance"
>
> How is the optimal ratio of synthetic to real data defined? Are you referring to a point where synthetic data provides maximum benefit, beyond which adding more may actually reduce accuracy?
>
> [1] Fan, Lijie, et al. "Scaling laws of synthetic images for model training... for now." Proceedings of the IEEE/CVF Conference on Computer Vision and Pattern Recognition. 2024.
>
> [2] Singh, Krishnakant, et al. "Is Synthetic Data all We Need? Benchmarking the Robustness of Models Trained with Synthetic Images." Proceedings of the IEEE/CVF Conference on Computer Vision and Pattern Recognition. 2024.

---

> > ### Comment · Reviewer_TBMi · 2024-11-14
> >
> > What I mean is that ImageNet-100 might not be sufficient to explain some scaling and generalization cases. Perhaps you could consider using ImageNet-1k instead. Additionally, I believe there should be an optimal ratio between synthetic data and generated data to achieve the best performance, although you may need to set more sample points for more solid experimental results.

---

### Official Review · Reviewer_4SSv · 2024-10-29

**Soundness:** 1
**Presentation:** 1
**Contribution:** 1
**Rating:** 3
**Confidence:** 3

**Summary:**

This paper explores the equivalence of generative and real data augmentation in classification

**Strengths:**

this paper try to figure out how fake data improve classification

**Weaknesses:**

1) the equations are weard. The authors try to give a precise formulation to model the effectiveness of generated data from large models. However, it's well-known that deep networks are too complex to be formulized into explicite equations.

2) Generated data do not provide new information, hence i do not trust this research topic. Although the authors mentioned external large models, the training of large models still need real data.

3) The final classification results are not competive to sota models like CoCa and ViT-e

**Questions:**

1) In euqtion 1, where is the number 15571.2 from?

2) could the authors provide reusults of Percentage correct on the  CIfar10 dataset?

---

### Official Review · Reviewer_AgTc · 2024-10-30

**Soundness:** 3
**Presentation:** 2
**Contribution:** 1
**Rating:** 3
**Confidence:** 3

**Summary:**

Paper investigate the synthesis data in image classification tasks. The authors compare real and synthetic data, generated by advanced text-to-image models, to assess the impact on classification performance across various scenarios, using datasets like CIFAR-10 and ImageNet. The study provides empirical guidelines for how much synthetic data is needed to match the performance benefits of real data augmentation. Results show that while synthetic data can improve classifier accuracy, especially in data-scarce environments, it requires larger volumes to achieve comparable performance with real data.

**Strengths:**

The paper rigorously quantifies the trade-offs between real and synthetic data, offering empirical formulas to guide the required synthetic-to-real data ratios for effective augmentation. This quantitative approach provides actionable insights for researchers and practitioners on optimizing data augmentation strategies.

By evaluating the impact of both internal (training set only) and external (large-scale pre-trained) synthetic data on well-known datasets (CIFAR-10, ImageNet), the study demonstrates a thorough exploration of generative data's utility.

**Weaknesses:**

The study's research question, while valuable, is already extensively explored within the synthetic data and image classification community. This paper does not introduce new insights or findings beyond what prior studies have already established. The results presented appear to align closely with existing, well-documented conclusions, providing minimal advancement in our understanding of synthetic data's impact on image classification.

1. Numerous papers cover synthetic data for image classification, including studies like arXiv:2304.08821, which also explores synthetic data's utility for classification tasks using modern generative models, arriving at similar conclusions. Additionally, arXiv:2204.08610 provides a comprehensive study of data augmentation across multiple classification datasets, reinforcing comparable outcomes, making the results here somewhat redundant.

2.  results in image classification does not translate well to other domains with higher current relevance, such as large language models (LLMs) or language-related synthetic data. Expanding the scope to more emergent areas—such as synthetic data for LLM training or even diffusion training—could yield broader contributions with wider applicability.

3 The main results in Table 1 primarily show a basic comparison between data size and performance gain in classification. However, as data size increases, factors like model size and architecture can significantly impact outcomes. Investigating performance variability across different model sizes or architectures, and including training variance across runs, would strengthen the findings and allow for a more nuanced analysis of the impact of synthetic data augmentation on classification performance.

4. it would be better to investigate more under-invesigated topics such as low-resource images, casual relations in image classification. Simply focusing on accuracy is not as meaningful for many research today as we already know "more high quality" data always leads to somewhat better results from all previous studies.

**Questions:**

For quality assessment, FID and IS are very bad scores, since the inception net is old and does not capture enough features. There are many new quality assesment metrics, such as Clip-MMD which uses modern clip model, does author take into consideration of some other scores?

---

> ### Author Response · Authors · 2024-11-14
>
> We would like to sincerely thank Reviewer AgTc for providing a detailed review and insightful questions.
>
> > W1:"Numerous papers cover synthetic data for image classification, including studies like arXiv:2304.08821 and arXiv:2204.08610"
>
> arXiv:2304.08821 proposes a new method for data augmentation. Although Figures 2 and 3 in their paper show accuracy curves for different synthetic image ratios under varying amounts of real images, their main focus is to validate the effectiveness of their method, rather than studying **the effectiveness of generative data augmentation for image classification on a given dataset**. In contrast, we provide a more comprehensive analysis, including a quantitative comparison between real and generative data augmentation, along with a diverse set of conclusions in Sections 3.1, 3.2, and 3.3. arXiv:2204.08610 includes generative data augmentation as one of the 10 augmentation techniques with results reported just on the mixed augmentation; it does not examine the effectiveness of generative data augmentation in particular. Please see also our reply to the common question.
> > W2:"Results in image classification does not translate well to other domains with higher current relevance, such as large language models (LLMs) or language-related synthetic data"
>
> Thank you for highlighting the area of LLMs. However, we feel this may not be highly relevant here, as pixels differ significantly from words. Focusing on conclusions that benefit the field of computer vision alone would already be a valuable contribution.
>
>
> > W3:"Factors like model size and architecture can significantly impact outcomes. Investigating performance variability across different model sizes or architectures, and including training variance across runs, would strengthen the findings."
>
> Thank you for the suggestions. Different classification models indeed affect performance, and we will work to include results from additional classifiers, though we believe they will have the similar trend. Including training variances is also a valuable suggestion. We did not report the variances, but we conducted the same experiments over three trials, and the results are robust.
>
> > W4:"Simply focusing on accuracy is not as meaningful for many research today as we already know "more high quality" data always leads to somewhat better results from all previous studies."
>
> Although "more high quality" leads to better quality seems to be trivial, we still consider studying the effectiveness of synthetic data worthwhile, because we need to take the computational and time cost into account, and choose whether to gather more real data or utilize generative models. The conclusions regarding the effectiveness of generative data augmentation provide valuable guidance to people on when and how to use synthetic data.
>
> > Q1:"For quality assessment, FID and IS are very bad scores. There are many new quality assessment metrics, such as Clip-MMD."
>
> Thanks for mentioning. Currently we have not used other quality assessment metrics, and we will consider them.

---

### Official Review · Reviewer_aYQx · 2024-10-31

**Soundness:** 3
**Presentation:** 3
**Contribution:** 1
**Rating:** 3
**Confidence:** 4

**Summary:**

The paper presents an empirical study to evaluate the effectiveness of generated data in image classification tasks. Specifically, the study reports the accuracy changes when training a classifier with various amounts of generated data added to CIFAR-10, ImageNet10, and ImageNet100 training datasets. The study reveals that (1) generated augmented data can improve classification accuracy, especially when real data is limited, (2) real data are more sample-efficient than synthesized data, and (3) the performance gains diminish when the amount of synthesized data increases.

**Strengths:**

- The analysis is clear and easy to understand.
- The study confirms the findings from previous works.

**Weaknesses:**

I appreciate the effort of the authors to systematically and comprehensively study the impact of using generated augmented data for classification tasks. The experiments are very clear and concise. However, most of the findings (1), (2), and (3) mentioned in the summary section above are generally known and have been reported from previous works [1] on more datasets. The proposed study works out the exact accuracy improvements concerning the syn-real ratios on three datasets. I am afraid this does not demonstrate sufficient novelty and technical contribution.

[1] Ruifei He, Shuyang Sun, Xin Yu, Chuhui Xue, Wenqing Zhang, Philip H. S. Torr, Song Bai, & Xiaojuan Qi (2023). Is Synthetic Data from Generative Models Ready for Image Recognition?. In The Eleventh International Conference on Learning Representations, ICLR 2023.

**Questions:**

Suggestion:

- The authors may consider working out some guidelines for selecting the optimal syn-real ratio for any new datasets. However, this probably needs more experiments on a larger variety of datasets.
- The current manuscript is more like a technical report. It may be more suitable to submit it to a journal related to machine learning applications.

---

> ### Author Response · Authors · 2024-11-14
>
> We would like to sincerely thank Reviewer aYQx for providing a detailed review and insightful questions.
> > W1： "Most of the findings are generally known and have been reported from previous works on more datasets".
>
> We admit that conclusion 1,2,3 are somewhat similar to [1], but the problem we investigate is much different than theirs. The differences can be summarized as follows:
>
> 1. Their work only focuses on few-shot learning, assuming that "the positive impact of synthetic data will gradually diminish as the amount of real data increases." However, there are two main issues with this approach.
> First, their conclusion relies simply only on observing a converging trend of accuracy curves with and without synthetic data when adding more real data. However, when a large amount of real data is available and the accuracy is already high, even if we add additional real data instead of synthetic, further accuracy improvements still become increasingly difficult. In other words, the positive impact of additional **real** data will also gradually diminish as the amount of given real data increases. Only by studying the **quantitative equivalence** of real and generative data augmentation can we reach a solid conclusion.
> Second, even if the positive impact of synthetic data does gradually diminish with more real data, it is still valuable to explore the improvements of synthetic data under this condition. Even a 0.x% accuracy improvement when accuracy is already high (e.g., around 90%) can be meaningful, as it could push results to a new SOTA level.
>
> 2. They consider only generative models trained on large-scale external datasets. However, in specialized scenarios—such as medical imaging—pre-trained generative models may perform poorly due to a significant distribution gap from common images. In these cases, internal data augmentation becomes essential to address the issue. Thus, the problem we investigate is **more foundational** than theirs.
>
> > Q1: "Guidelines for selecting the optimal syn-real ratio for any new datasets"
>
> This is a good question, but could you provide a more concrete definition of the optimal syn-real ratio? Is it a point where synthetic data provides maximum benefit, beyond which adding more may actually reduce accuracy?
>
> > Q2: "The current manuscript is more like a technical report."
>
> We respectively disagree with this comment. As mentioned above, our work on the quantitative equivalence of real v.s. synthetic data and internal v.s. external augmentation are foundational and helpful to the machine-learning community. We provide insightful guidance for the future research on synthetic data. Please also see our reply to the common questions at the beginning.
>
>
> [1] Ruifei He, Shuyang Sun, Xin Yu, Chuhui Xue, Wenqing Zhang, Philip H. S. Torr, Song Bai, & Xiaojuan Qi (2023). Is Synthetic Data from Generative Models Ready for Image Recognition?. In The Eleventh International Conference on Learning Representations, ICLR 2023.

---

> > ### Comment · Reviewer_aYQx · 2024-11-14
> >
> > Optimal syn-real ratio. There can be multiple definitions of the optimal syn-real ratio. Achieving maximum accuracy can be one definition; efficacy can be another. In a more general case, how can your work help a data practitioner select a good syn-real ratio if encountering new datasets in practice?

---

### Official Review · Reviewer_gL25 · 2024-11-01

**Soundness:** 2
**Presentation:** 2
**Contribution:** 2
**Rating:** 5
**Confidence:** 4

**Summary:**

This paper is an empirical study on the usage of generated data (through generative modelling) for classifier training. The authors introduce a framework in which they analyze the effectiveness on adding generated data in the training. They have two setup: 1) In the first one they train a generative model on their target dataset and investigate how generating additional data improve the classifier's performance. 2) In the second setup, they leverage off-the-shelve large generative models such as StableDiffusion to generate additional data. One of their main takeaway is that when having low amount of training data, then augmenting the training set with generated data improve the performances. However, the gain are less significant when having larger amount of real data. Another main take away is that generated images are much more less "data efficient" than real images, so there is a need to increase significantly the number of generated point.

**Strengths:**

- The paper is well written and easy to follow. I really appreciate the main result highlights.
- The authors performed a in depth empirical analysis of what is the benefit of synthetic data for classifier training.
- The experimental setup on CIFAR10 and ImageNet is convincing.
- Lot of details about the experimental setup and the implementations.
- Valuable ablation study over the impact of the ratio of synthetic data used.

**Weaknesses:**

- I would be cautious about such strong statement as "synthetic data is less sample efficient than real data". It might just be because of the way those samples are generated or the lack of diversity in the generated sample. As highlighted by the authors, there is a significant domain gap between the generated and real images, so I am not sure that we can say that synthetic data is per say less efficient, but maybe, we could say that current generative models, even if they are able to generate visual appealing images, they are still not able to match the distribution of natural images. And it happens that this failure of current generative model, induces learning a distribution that is indeed less diverse and real than the natural images which make then less useful for sampling relevant data for classifier training.
- This paper provide mostly an empirical study that has indeed value bit I would have expected to see a more fine-grained analysis such as analyzing wether the ratio between real/synthetic data has the same impact across all classes, or if some classes benefits more than others.

**Questions:**

- Do you think that this lack of "efficiency" from synthetic data might due to the fact that those generative models are optimized for quality and not so much for diversity? Any other idea on why this could be happening?
- I am wondering if you have any suggestions on how to improve the "efficiency" of the synthetic data?

---

> ### Author Response · Authors · 2024-11-14
>
> We would like to sincerely thank Reviewer gL25 for providing a detailed review and insightful questions.
> > W1: "cautious about such strong statement as 'synthetic data is less sample efficient than real data'"
>
> We sincerely thank Reviewer gL25 for the thoughtful and careful statement. We will adjust the presentations in our paper.
> > W2:"'synthetic data is less sample efficient than real data' might just because the generated samples lack diversity."
>
> **Synthetic data is less sample-efficient than real data, not only due to limited diversity but also because of fundamental differences in their underlying manifolds.**
>
> The diversity of synthetic data indeed plays a critical role in classifier accuracy. As demonstrated in Figure 4, Sable Diffusion 3 images, with greater diversity than Stable Diffusion 2, lead to a higher accuracy boost. However, as both the paper and reviewer gL25 highlight, a substantial domain gap persists between generated and real images, preventing synthetic data from fully matching the distribution of natural images. Regardless of generation methods, synthetic data consistently diverges from real data. Even in internal augmentation settings without external data introducing a **data domain gap**, synthetic data generated through different models and techniques still fails to outperform real data.
>
> > Q1: "a more fine-grained analysis such as analyzing whether the ratio between real/synthetic data has the same impact across all classes or if some classes benefit more than others."
>
> This is a great question and worthy to be studied in the follow-up work. However, it is beyond the scope of this work. We will consider this in the future work.
>
> > Q2: "any suggestions on how to improve the "efficiency" of the synthetic data?"
>
> We tried multiple ways to improve the efficiency of the synthetic data.
> 1. We first developed an edge decoder to extract edge information from mid-layer features of real or synthetic data, as we found that distinguishing edges between real and synthetic images was more challenging than separating the original images themselves.
> 2. Treating real and synthetic data as separate domains, we tried using domain adversarial training techniques to close the gap between real and synthetic features.
> 3. We also tried extracting edges from real images and generating synthetic data using a ControlNet with the extracted edges as the control condition.
>
>
> However, methods 1 and 2 do not yield better results than simply mixing real and synthetic data, while method 3 struggles to generate high-fidelity images, as the edges of real images in ImageNet are typically too noisy to provide a reliable generation condition.

---

### Official Review · Reviewer_gJjx · 2024-11-02

**Soundness:** 3
**Presentation:** 3
**Contribution:** 3
**Rating:** 6
**Confidence:** 3

**Summary:**

The paper examines the impact of generative data augmentation in image classification tasks. It offers a comparison between real and synthetic data, with the latter being produced by advanced text-to-image models, and discusses the benefits of using synthetic data to improve classification performance. The study demonstrates that generative data augmentation can significantly enhance classification performance, especially when real data is scarce, and provides empirical guidelines for determining the appropriate size of synthetic dataset augmentation. The results on CIFAR-10 and ImageNet highlight the influence of the baseline training set size and the quality of generative models on the effectiveness of classification.

**Strengths:**

S1: This study presents a lucid and accessible exploration of the impact of generative data augmentation in the domain of image categorization. It offers a thorough examination of the efficacy of generative data enhancement, not only assessing the performance of generative models across various datasets but also dissecting the interplay between synthetic and real data. The research is bolstered by an extensive array of experimental results, leading to direct and tangible conclusions that enhance the reader's comprehension.

S2: Extensive experimentation substantiates the efficacy of generative data augmentation, particularly under conditions where real data is constrained. The study demonstrates that synthetic data can markedly enhance classification performance, offering valuable insights for practical applications and contributing to the body of knowledge in the field.

S3: The study's methodology is rigorous, employing several renowned datasets and well-established generative models to conduct experiments. This approach ensures the reliability and generalizability of the findings, thereby reinforcing the study's scholarly contribution to the field of image classification and data augmentation.

**Weaknesses:**

W1: There are shortcomings in the ablation studies, as they do not adequately analyze the specific impact of different generative models or the effect of varying dataset sizes. It would be beneficial to compare performance across a wider range of generative model architectures and to systematically analyze how results change with different synthetic dataset sizes. Such comparisons would provide clearer insights into the strengths and limitations of the methods used.

W2 : The paper does not discuss the significant computational resources required for generating high-quality synthetic data. To enhance clarity, I recommend that the authors provide specific details about the computational overhead, including GPU hours, memory requirements, and energy consumption. This information would help readers understand the practical implications of implementing this approach.

W3: Real-world scenarios with variable conditions, such as those involving autonomous vehicles, may not have adequate validation

**Questions:**

1. Why were generative models like Vanilla-DDPM and Stable Diffusion 2 chosen for image generation? How might the choice of these specific models impact the generalizability of the results? Are there alternative models that could also be explored in future work?

2. Can you clarify the rationale behind selecting different image encoders at various stages of the generation process, such as CLIP and DINOv2? How does this choice influence the final results?

---

> ### Author Response · Authors · 2024-11-14
>
> We would like to sincerely thank Reviewer gJjx for providing a detailed review and insightful questions.
>
> > W1: "There are shortcomings in the ablation studies, as they do not adequately analyze the specific impact of different generative models or the effect of varying dataset sizes."
>
> In fact, we systematically explored the effects of different generative models and dataset sizes across CIFAR-10, ImageNet-10, ImageNet-100 datasets in our experiments.
>
> For CIFAR-10, we utilized two generative models trained on internal dataset: Vanilla-DDPM and EDM. Additionally, we used the CIFAKE dataset generated by Stable-Diffusion-1.4 and resized to 32x32. To examine the impact of dataset size, we tested with three scales of real data (500, 2500, and 5000 samples per class) along with varying proportions of synthetic data.
>
> On ImageNet-10 and ImageNet-100, we experimented with Stable Diffusion 2 and Stable Diffusion 3 using diverse prompts to generate synthetic data. Real dataset sizes included 65, 260, and 1300 samples per class, again with varying levels of synthetic data integration.
>
> > W2: "The paper does not discuss the significant computational resources required for generating high-quality synthetic data."
>
> Thanks for mentioning the aspect of computational resources. Here, we provide the memory usage and GPU hours for the generative models used in our experiments:
>
> 1. CIFAR-10 Experiments (A4500 GPU)
> - **EDM**: 18 sampling steps, batch size of 100
>   - Memory usage: 4000MB
>   - Time per image: 0.06 seconds
>
> - **Vanilla-DDPM**: 500 sampling steps, batch size of 100
>   - Memory usage: 3300MB
>   - Time per image: 0.95 seconds
>
> We used 4 A4500 GPU, and it took 16 hours to generate Vanilla-DDPM data at 5 times the scale of CIFAR-10.
>
> 2. ImageNet Experiments (A6000 GPU)
> - **Stable Diffusion 2**: 50 sampling steps
>   - Memory usage: 3880MB
>   - Time per image: ~1 second
>
> - **Stable Diffusion 3**: 28 sampling steps
>   - Memory usage: 21808MB
>   - Time per image: ~7 seconds
>
> We used 4 A6000 GPUs, and it took 2~3 days to generate SD-3 synthetic data at the same scale as ImageNet-100.
>
> > W3: Real-world scenarios with variable conditions, such as those involving autonomous vehicles, may not have adequate validation
>
> We agree that the impact of synthetic data can be further studied on more specific real-world tasks. However, this paper focuses on identifying general trends, and in future work, we plan to validate our findings on tasks like autonomous driving or medical imaging.
>
> > Q1: Why were generative models like Vanilla-DDPM and Stable Diffusion 2 chosen for image generation? How might the choice of these specific models impact the generalizability of the results? Are there alternative models that could also be explored in future work?
>
> We selected Vanilla-DDPM and Stable Diffusion 2 due to their widespread use for generating synthetic data. We acknowledge that various generative models could affect classification outcomes, yet we believe the core conclusions of our paper remain robust. Our findings with EDM and Stable Diffusion 3 support this view, and we leave the exploration of additional generative models to future work. There are many alternatives to explore, like GLIDE, Imagen, Muse, and DALL-E for diffusion-based models. For further details, please refer to the "Text-to-Image Diffusion Models" section in "4 Related Work".
>
> > Q2: The rationale behind selecting different image encoders at various stages of the generation process, and the influence to the final results.
>
> For external generative data augmentation, we use the original Stable Diffusion configurations. They use CLIP as the **text encoder** because CLIP acts as a "translator" between language and visuals, allowing Stable Diffusion to create images that match the text prompt closely. DINOv2 is not used in the whole process. For internal generative data augmentation, we use ordinary conditional data generation without text prompts.

---

### Note · Authors · 2024-11-15

I have read and agree with the venue's withdrawal policy on behalf of myself and my co-authors.